# Fast and Accurate Lane Detection via Graph Structure and Disentangled Representation Learning

**DOI:** 10.3390/s21144657

**Published:** 2021-07-07

**Authors:** Yulin He, Wei Chen, Chen Li, Xin Luo, Libo Huang

**Affiliations:** College of Computer, National University of Defense Technology, Changsha 410073, China; heyulin@nudt.edu.cn (Y.H.); lichen14@nudt.edu.cn (C.L.); luoxin13@nudt.edu.cn (X.L.); libohuang@nudt.edu.cn (L.H.)

**Keywords:** lane detection, graph structure, feature compression, disentangled representation learning

## Abstract

It is desirable to maintain high accuracy and runtime efficiency at the same time in lane detection. However, due to the long and thin properties of lanes, extracting features with both strong discrimination and perception abilities needs a huge amount of calculation, which seriously slows down the running speed. Therefore, we design a more efficient way to extract the features of lanes, including two phases: (1) Local feature extraction, which sets a series of predefined anchor lines, and extracts the local features through their locations. (2) Global feature aggregation, which treats local features as the nodes of the graph, and builds a fully connected graph by adaptively learning the distance between nodes, the global feature can be aggregated through weighted summing finally. Another problem that limits the performance is the information loss in feature compression, mainly due to the huge dimensional gap, e.g., from 512 to 8. To handle this issue, we propose a feature compression module based on decoupling representation learning. This module can effectively learn the statistical information and spatial relationships between features. After that, redundancy is greatly reduced and more critical information is retained. Extensional experimental results show that our proposed method is both fast and accurate. On the Tusimple and CULane benchmarks, with a running speed of 248 FPS, F1 values of 96.81% and 75.49% were achieved, respectively.

## 1. Introduction

In the past decade years, automatic driving has gained much attention with the development of deep learning. As an essential perception task in computer vision, lane detection has long been the core of automatic driving [1]. Despite the long-term research, lane detection still has the following difficulties: (1) Lanes are slender curves, the local features of them are more difficult to extract than ordinary detection tasks, e.g., pedestrians and vehicles. (2) Occlusion is serious in lane detection so that there are few traceable visual clues, which requires global features with long-distance perception capabilities. (3) The road scenes are complex and changeable, which puts forward high requirements for the real-time and generalization abilities of lane detection. Figure 1 shows the realistic lane detection scenes under occlusion, illumination change, strong exposure, and night conditions.

Traditional lane detection methods usually rely on hand-crafted features [2,3,4,5,6,7,8], and fit the lanes by post-processing, e.g., Hough transforms [2,3]. However, traditional methods require a sophisticated feature engineering process, and cannot maintain robustness in real scene, hindering their applications.

With the development of deep learning, a large number of lane detection methods based on convolutional neural networks (CNN) have been proposed [9,10,11,12,13], which greatly improves the performance. The mainstream lane detection methods are based on segmentation which predict the locations of lanes by pixel-wise classification with an encoder-decoder framework. They first utilize a specific backbone (composed with CNN) as the encoder to generate feature maps from the original image, then use an up-sampling module as the decoder to enlarge the size of feature maps, performing a pixel-wise prediction. However, the lanes are represented as segmented binary features in segmentation methods, which makes it difficult to aggregate the overall information of lanes. Although some works [10,12,14] utilize specially-designedspatial feature aggregation modules to effectively enhance the long-distance perception ability. However, they also increase the computational complexity and make the running speed slower. Moreover, most segmentation-based methods need to use post-processing operations (e.g., clustering) to group the pixel-wise predictions, which is also time-consuming.

In order to avoid the above-mentioned shortcomings of segmentation-based methods, a great number of works [15,16,17,18] began to focus on using different modeling methods to deal with the lane detection problem. Polynomial-based methods [16,18] propose to localize lanes by learnable polynomials. They project the real 3D space into 2D image space and fit a series of point sets to determine the specific coefficients. Row-based classification methods [15,19] detect lanes by row-wise classification based on a grid division of the input image. Anchor-based approaches [17,20] generate a great number of anchor points [20] or anchor lines [17] in images and detect lanes by classifying and regressing them. The above methods consider the strong shape prior of lanes and can extract the local features of lanes more efficiently. Besides, these works discard the heavy decoding network and directly process the high-dimensional features generated by the encoding network, so the real-time ability of them is stronger than segmentation-based methods. However, for fewer calculation and training parameters, most methods above directly use 1 × 1 convolution to compress high-dimensional features to complete downstream tasks (i.e., classification and regression tasks). Because the dimension difference between input features and compressed features is too large, the information loss problem is serious in the feature compression, which affects the upper bound of accuracy.

Consider the existing difficulties and the recent development of lane detection, we propose a fast and accurate method to detect lanes, namely FANet, aiming to resolve the three difficulties mentioned in the first paragraph. For the first issue, we utilize a line proposal unit (LPU) to effectively extract the local lane feature with strong discrimination. LPU generates a series of anchor lines over the image space and extracts the thin and long lane features by the locations of anchor lines. For the second problem, we propose a graph-based global feature aggregation module (GGFA) to aggregate global lane features with strong perception. GGFA treats local lane features as nodes of the graph, and adaptively learn the distances between nodes, then utilizes weighted sums to generate global feature. This graph is fully connected, its edges represent the relations between nodes. GGFA can effectively capture visual cues and generate global features with strong perception ability. For the third difficulty, to pursue higher running speed, we also drop the decoder network and compress the high-dimensional feature map as above works. However, unlike they directly use 1 × 1 convolution to compress features, we utilize the idea of disentangled representation learning [21,22] to restain more information in compressed features and name this module as Disentangled Feature Compressor (DFC). Specifically, DFC divides the high-dimensional feature map into three groups and integrates features respectively through low-dimensional 1 × 1 convolution. Then, inspired by batch normalization (BN) operation [23], DFC follows the “normalize and denormalize” steps to learn the statistical information in the spatial dimension, so that the representation of the compressed feature will be richer. Moreover, sufficient experimental results in different scenarios also prove the strong generalization ability of our method.

Extensive experiments are conducted on two popular benchmarks, i.e., Tusimple [24] and CULane [10], our proposed FANet achieves higher efficacy and efficiency compared with current state-of-the-art methods. In summary, our main contributions are:We propose a fast and accurate lane detection method, which aims to alleviate the main difficulties among lane detection problems. Our method achieves state-of-the-art performance on Tusimple and CULane benchmarks. Besides, the generalization of it is also outstanding in different driving scenarios.We propose an efficient and effective global feature aggregator, namely GGFA, which can generate the global lane feature with strong perception. This module is a general module that can apply to other methods whose local features are available.We propose a general feature compressor based on disentangled representation learning, namely DFC, which can restrain more information in the compressed feature without speed delay. This module is suitable to feature compression with huge dimensional differences, which can greatly improve the upper bound of accuracy.

## 2. Related Work

Lane detection is a basic perceptual task in computer vision and has long been the core of autonomous driving. Due to scene difficulties such as occlusion and illumination changes, as well as realistic requirements such as generalization and real-time, lane detection is still a challenging task. The current deep learning methods can divide into four categories: segmentation-based methods, anchor-based methods, row-based methods, and polynomial-based methods.

### 2.1. Segmentation-Based Methods

Segmentation-based methods are the most common manner to detect lanes and have achieved significant success. They locate the positions of lanes by predicting the pixel-level categories of the image. Different from general segmentation tasks, lane detection needs instance-level discrimination, somewhat like an instance segmentation problem. Some methods proposed to utilize multi-class classification to resolve this problem, yet they can only detect a fixed number of lanes. For higher flexibility and accuracy, some methods add a post-clustering strategy to group the lanes. However, this post-process is always time-consuming. Another problem that affects accuracy is that lane detection needs a stronger receptive field than general segmentation tasks due to the long and thin structure of lanes. Bell et al. [25] utilized recurrent neural networks (RNN) to transmit the horizontal and vertical contextual information in the image to improve the spatial structure perception of features. Liang et al. [26] constructed a graph long short-term memory layer (LSTM) to provide a direct connection for spatial long-distance information transmission and enhance the ability of long-distance perception. Pan et al. [10] proposed a specifically designed scheme for long and thin structures and demonstrated its effectiveness in lane detection. However, the above operations are time-consuming due to the long-distance information communication. Recently, some studies [19,27] indicated that it is inefficient to describe the lane as a mask because segmentation-based methods can not emphasize the shape prior of lanes. To overcome this problem, row-based, polynomial-based, and anchor-based methods are proposed.

### 2.2. Row-Based Methods

Row-based methods have a good use of shape prior of lanes and predict locations of lanes by the classification of each row. They fully utilize the characteristic that lane lines do not intersect horizontally and add the location constraint of each row to achieve the continuity and consistency of lanes [19,28]. Besides, some recent row-wise detection methods [15,19] have achieved advantages in terms of efficiency. But as the widely used post-clustering module [29] in segmentation-based methods cannot be directly integrated into the row-wise manner, row-wise methods can only detect fixed lanes by multi-class classification strategy.

### 2.3. Polynomial-Based Methods

Polynomial-based methods utilize learnable polynomials on image space to fit the locations of lanes. PolyLaneNet [16] first proposed to localize lanes by regressing the lane curve equation. LSTR [18] introduced transformer [30] and Hungarian Algorithm to achieve a fast end-to-end lane detection method. However, polynomial-based methods have not surpassed other methods in terms of accuracy. Besides, they usually have a high bias towards straight lanes in their predictions.

### 2.4. Anchor-Based Methods

Inspired by the idea of anchor-based detection methods like [31,32], anchor-based lane detection methods are proposed to generate a great number of anchors. Due to the slender shape of lane lines, the widely used anchor boxes in object detection cannot be used directly. Anchor-based lane detection methods usually define a large number of anchor points [20] or anchor lines [17] to extract local lane feature efficiently, then classify them into certain categories and regress the relative coordinates.

## 3. Methods

We first introduce the overall architecture of our proposed FANet in Section 3.1. Then, the DFC module, a feature compressor that is based on disentangled representation learning, will be introduced in Section 3.2. Subsequently, Line Proposal Unit (LPU), an anchor lines generator will be introduced in Section 3.3. The proposed GGFA module, an effective and efficient global lane feature aggregator, will be introduced in Section 3.4. Finally, we elaborate on the details of the model training process in Section 3.5.

### 3.1. Architecture

FANet is a single-stage anchor-based detection model (like YOLOv3 [31] or SSD [32]) for lane detection. The overview of our method is as shown in Figure 2.

It receives an RGB image $I\in R^{H_{I}\times W_{I}\times 3}$ as input, which is taken from the camera mounted in a vehicle. Then, an encoder (such as ResNet [33]) extracts the features of *I*, outputting a high-dimensional feature map $F_{H}\in R^{H_{F}\times W_{F}\times C}$ with deep semantic information, where $H_{F}$, $W_{F}$, and *C* are the height, the width, and the channel dimension of $F_{H}$. For fast running, the DFC module we proposed is then applied into $F_{H}$ and generates compressed feature $F_{C}\in R^{H_{F}\times W_{F}\times \hat{C}}$ with a low dimension, where $\hat{C}$ is the channel dimension of $F_{C}$. After that, LPU generates predefined anchor lines which are throughout the image and extracts local lane features by their locations from $F_{C}$. To enhance the perception of the features, GGFA builds a graph structure and aggregates global lane features with the input of local lane features. Then, we concatenate the local lane features and the global features, then predict the lanes by two fully connected layers (FC). The first one is to classify the proposal lanes are background or targets. The second one is to regress relative coordinates. Since the lane is represented by 2D-points with fixed equally-spaced *y*-coordinates, the corresponding *x*-coordinates and the length of the lane are the targets in the regression branch.

### 3.2. Disentangled Feature Compressor

Our proposed DFC aims to preserve more information of compressed features, thereby improving the representation of features. The core of it is to exploit the idea of disentangled representation learning to reduce the correlation of feature components. Besides, the module design is inspired by the classic batch normalization (BN) operation, which follows the “normalize and denormalize” steps to learn the feature distribution. Next, we will detail the structure design, the theory, and the computational complexity of our module.

#### 3.2.1. Structure Design

As shown in Figure 3, DFC module receives a high-dimensional feature $F_{H}$ as input, the size of it is [H,W,C]. Then, DFC divides $F_{H}$ into three groups and generates three low-dimensional feature maps $F_{L}^{1},F_{L}^{2},F_{L}^{3}$ through three 1 × 1 convolutions that do not share weights. The size of the low-dimensional feature map is $\left[H,W,\hat{C}\right]$. After that, normalization operation in the spatial dimension is performed onto $F_{L}^{1}$, making the features more compact. Finally, with the input of $N(F_{L}^{1})$, $F_{L}^{2}$, and $F_{L}^{3}$, denormalization operation consisting of element-wise product and element-wise add is then used to learn the spatial statistical information, thus enhancing the diversity of the compressed feature.

#### 3.2.2. Theory Analysis

As mentioned above, the core of DFC is based on decoupling representation learning, which can reduce the coupling between features, thereby enhancing feature diversity. Different from common decoupling representation learning tasks [34,35,36] that branches have different supervision goals respectively. In our module, the three branches are all supervised by the final targets, i.e., classification and regression tasks. Nevertheless, the “divide and conquer” strategy of decoupling representation learning is exploited in our module. After deciding the main idea of our module, we make an assumption for the feature compression problem, i.e., the high-dimensional feature has a great amount of redundant information, and suitable division does not have much impact on its representation ability. Therefore, we divide the high-dimensional feature map into three components, each of which has a different effect on the target.

Inspired by the classic batch normalization algorithm, we apply the “normalize and denormalize” steps to learn the statistical information in the feature plane. The formulation of BN is as follows:(1)$${\tilde{Z}}^{\left[l\right]}=\gamma \cdot \frac{Z^{\left[l\right]}-\mu }{\sqrt{\sigma^{2}+\epsilon }}+\beta$$
where γ and β are the learnable parameters to perform the “denormalize” operation. $Z^{\left[l\right]}$ denotes the *l*-th sample in a mini-batch. μ and σ are the mean and standard deviation of *Z*, ϵ is a small number, preventing the denominator from being zero.

Instead of learning the mini-batch sample distribution in BN, we propose to learn the spatial feature distribution. The computation is as follows:(2)$${\tilde{F}}_{L}=F_{L}^{2}\left(\frac{F_{L}^{1}-\mu_{s}\left(F_{L}^{1}\right)}{\sigma_{s}\left(F_{L}^{1}\right)\;+\;\epsilon }\right)\;+\;F_{L}^{3}$$
where $\mu_{s}$ and $\sigma_{s}$ are the mean and standard deviation of $F_{L}^{1}$ in the spatial feature dimension. $F_{L}^{2}$ and $F_{L}^{3}$ are analogous to the γ and β in Equation (Equation 1), which are also learnable. Therefore, the three independent components, i.e., $F_{L}^{1}$. $F_{L}^{2}$, and $F_{L}^{3}$ all have the different functions towards the target. $F_{L}^{1}$ is the component to control the specific value, somewhat like the “value” component of transformer [30]. $F_{L}^{2}$ controls the deviation of the main distribution, and also decides the preservation degree of $F_{L}^{1}$. $F_{L}^{3}$ controls the bias of the distribution.

#### 3.2.3. Computational Complexity

We compare the computational complexity of the 1 × 1 convolution and the DFC module we designed in this section. The computational complexity of 1 × 1 convolution is as follows:(3)$$S_{c}=1\times 1\times H\times W\times C\times \hat{C}.$$

While the computational complexity of DFC is written as:(4)$$S_{d}=S_{c}+O\left(H\times W\times \hat{C}\right)$$
where the convolution part of two operations is the same, while DFC adds normalization operation, element-wise product, and element-wise add. The extra computational complexity is far less than $S_{c}$. Taking C=512 and $\hat{C}=64$ as an example, calculation increment is only 1%, which can almost be ignorable.

### 3.3. Line Proposal Unit

The structure of LPU is as shown in Figure 2, which generates a series of anchor lines and extracts local lane features by their locations. In this way, these local features have a strong shape prior, i.e., thin and long, thereby having a stronger discrimination ability.

As shown in Figure 4, we define the anchor line as a straight ray. LPU generates lines from the left, bottom, and right boundaries. A straight ray is set with a certain orientation θ and each starting point $\left\{X_{s},Y_{s}\right\}$ is associated with a group of rays. However, the number of anchor lines greatly influences the efficiency of the method. Therefore, we exploit the statistical method to find the most common locations of lanes, thereby reducing the number of anchor lines. Specifically, we first approximate the curve lanes as straight lines in the training samples and record the angles and the start coordinates of them. Then, we utilize K-means clustering algorithm to find the most commonly used clusters, thereby getting the set of anchor lines. Table 1 shows the results after clustering representative angles. After getting the set of anchor lines, LPU can extract the feature of each anchor line by its location, which is composed of many pixel-level feature vectors. To ensure that the feature dimension of each anchor line is the same, we uniformly sample pixels in the height dimension of the feature map $y_{i}=\left\{0,1,2,\ldots ,H_{F}-1\right\}$. The corresponding *x*-coordinates can be obtained by a projection function:(5)$$\begin{array}{c} x_{i}=\left\lfloor \frac{1}{\tan\theta }\left(y_{i}-Y_{s}/S\right)+X_{s}/S\right\rfloor \end{array}$$
where *S* is the global stride of the backbone. Then, we can extract the local feature $F_{loc}=\left\{F_{loc}^{1},F_{loc}^{2},\ldots ,F_{loc}^{H_{F}-1}\right\}$ for each anchor line based on projected coordinates. In cases where $\left\{x_{i},y_{i}\right\}$ is outside the boundaries of $F_{C}$, $F_{loc}^{i}$ is zero-padded. Finally, the local lane features $F_{loc}\in R^{H_{F}\times \hat{C}}$ of anchor lines with the same dimension can be obtained.

### 3.4. Graph-Based Global Feature Aggregator

Lane detection requires a strong perception to locate the positions of lanes, but the local features cannot effectively perceive the global structure of the image, thus we propose GGFA to extract global lane features based on graph structure.

The structure of GGFA is as shown in Figure 5. It receives the local lane features $F_{loc}\in R^{H_{F}\times \hat{C}}$ as input and outputs the global lane features $F_{glb}\in R^{H_{F}\times \hat{C}}$, which have the same dimension as $F_{loc}$. Specifically, with the input of $F_{loc}$, Multi-Layer Perceptron (MLP) generates a distance vector with $\frac{N^{2}-N}{2}$ dimensions. Then, the distance vector is filled in the upper half part of an all-zero matrix. Flipping operation is applied to this matrix, making it perform as an axisymmetric matrix. In this way, the graph constructed by local features is a fully connected undirected graph. The distance between two nodes is the same for each one. After that, the weight matrix can be got as follows:(6)$$W_{i,j}=1-\underset{\mathrm{Softmax}}{\underbrace{\frac{e^{D_{i,j}}}{\sum_{j}e^{D_{i,j}}}}}$$
where $D_{i,j}$ is the distance between *i*-th and *j*-th nodes. Softmax operation transforms the distances to a soft value, then 1 reduces this soft value, which represents the similarity of two nodes. The distance between the two nodes is closer, the similarity weight will be bigger. After performing matrix transpose and element-wise product operation, the global feature $F_{glb}$ can be obtained finally.

This module learns the relationship between the local features by constructing the graph structure. Because the anchor lines are all over the whole image, the learned global features also fully consider the spatial relationship and visual cues. Therefore, the local features with long-distance can also be effectively communicated, thus enhancing the perception of features. At the same time, using adaptive weighted summation, the importance of each local feature can also be distinguished, making the learned global features informative.

### 3.5. Module Training

Similar to object detection, anchor-based lane detection methods also need to define a function to measure the distance between two lanes. For two lanes with common valid indices (i.e., equal-distance *y*-coordinates), the *x*-coordinates are $X_{a}={\left\{x_{i}^{a}\right\}}_{i=1}^{N_{pts}}$ and $X_{b}={\left\{x_{i}^{b}\right\}}_{i=1}^{N_{pts}}$, respectively, where $N_{pts}$ is the number of common points. The lane distance metric proposed in [17] is adopted to compute the distance between two lanes:(7)$$\begin{array}{c} D\left(X_{a},X_{b}\right)\;=\;\left\{\begin{array}{cc} \frac{1}{e^{\prime }-s^{\prime }+1}\cdot \sum_{i=s^{\prime }}^{e^{\prime }}\left|x_{i}^{a}-x_{i}^{b}\right|, & e^{\prime }\geq s^{\prime }\\ +\infty , & \;\mathrm{else}\; \end{array}\right. \end{array}$$
where $s_{a}$ and $s_{b}$ are the start valid indices of two lanes, $e_{a}$ and $e_{b}$ are the end valid indices of two lanes, and $s^{\prime }=max\left(s_{a},s_{b}\right)$ and $e^{\prime }=min\left(e_{a},e_{b}\right)$ define the range of those common indices.

Based on the distance metric in Equation (Equation 7), the process of training sample assignment can be defined. We compute the distance between anchor lines and targets, the anchor lines with a distance lower than a threshold $\tau_{p}$ are considered as positive samples, while those with a distance larger than a threshold $\tau_{n}$ are considered as negative samples.

The final loss function consists of two components, i.e., classification loss and regression loss, which are implemented by Focal loss [37] and L1 loss, respectively. The total loss can be defined as:(8)$$\begin{array}{cc} L\left({\left\{p_{i},r_{i}\right\}}_{i=0}^{N_{p\&n}-1}\right)\;= & \lambda \sum_{i}L_{cls}\left(p_{i},p_{i}^{*}\right)\\ \; & +\sum_{i}L_{reg}\left(r_{i},r_{i}^{*}\right) \end{array}$$
where $N_{p\&n}$ is the number of positive and negative samples, λ is used to balance the loss terms, $p_{i}$, $r_{i}$ are the classification and regression predictions of the *i*-th anchor line, and $p_{i}^{*}$, $r_{i}^{*}$ are the corresponding classification and regression targets. $p_{i}^{*}$ consists of “0” and “1”, i.e., background and lanes. $r_{i}^{*}$ is composed with the length *l* and the *x*-coordinates.

## 4. Experiments

In this section, we first introduce the datasets of our experiments, i.e., Tusimple [24] and CULane [10] benchmarks in Section 4.1, then present the implementation details of our methods in Section 4.2. After that, we compare the performance of our proposed FANet with other SOTA methods in Section 4.3 and conduct sufficient ablation studies to prove the effectiveness of our proposed modules in Section 4.4.

### 4.1. Dataset

Tusimple [24] and CULane [10] are the most popular benchmarks in lane detection. The TuSimple dataset is collected with stable lighting conditions in highways. The CULane dataset consists of nine different scenarios, including normal, crowd, dazzle, shadow, no line, arrow, curve, cross, and night in urban areas. More details about these datasets can be seen in Table 2.

#### 4.1.1. TuSimple

TuSimple is a lane detection dataset for highway scenes, which is used for the primary evaluation of lane detection methods. This dataset contains 3626 training images and 2782 test images. The image size in TuSimple is 1280×720, and each image contains up to 5 lanes.

On TuSimple, the main evaluation metric is accuracy, which is computed as:(9)$$accuracy=\frac{\sum_{clip}C_{clip}}{\sum_{clip}S_{clip}}$$
where $C_{clip}$ is the number of lane points that are predicted correctly and $S_{clip}$ is the total number of evaluation points in each clip. For each evaluation point, if the predicted point and the target point are within 20-pixel values, the prediction is considered to be correct, otherwise wrong. Moreover, we also calculate the false-positive rate (FP), the false-negative rate (FN), and the F1 score on predictions.

#### 4.1.2. CULane

CULane is a large lane detection dataset containing multiple scenarios. CULane has 98,555 training images and 34,680 test images. The image size is 1640×590, and each image contains up to 4 lanes.

The evaluation metric of CULane benchmark is F1 score, which can be defined as:(10)$$F_{1}=\frac{2\times Precision\times Recall}{Precision+Recall}$$
where $Precision=\frac{TP}{TP+FP}$ and $Recall=\frac{TP}{TP+FN}$. Different from TuSimple, each lane is considered as a line with a width of 30 pixels. Intersection-over-union (IoU) is calculated between predictions and targets. Those predictions with IoUs larger than a threshold (e.g., 0.5) are considered as correct.

### 4.2. Implementation Details

For all datasets, the input images are resized to 360 × 640 by bilinear interpolation during training and testing. Then, we utilize a random affine transformation (with translation, rotation, and scaling) along with random horizontal flips for data augmentaion. Adam [38] is adopted as the optimizer for training, the epoch is 100 in Tusimple dataset and 15 in CULane dataset. Learning rate is set to 0.003 with the *CosineAnnealingLR* learning rate schedule. To ensure the consistency of the experimental environment, all experimental results and speed measurements are performed on a single RTX 2080 Ti GPU. The number of anchor lines is set to 1000, the number of evaluation points ($N_{pts}$) is 72, the threshold for positive samples ($\tau_{p}$) is set to 15, and the threshold for negative samples ($\tau_{n}$) is set to 20.

To ensure the invisibility of the test dataset during the training process, we divide a small part of the two datasets as validation datasets for saving the optimal model. 358 and 9675 images were selected in TuSimple and CULane datasets, respectively.

### 4.3. Comparisons with State-of-the-Art Methods

In this section, we compare our FANet with other state-of-the-art methods on TuSimple and CULane benchmarks. In addition to the F1 score and accuracy, we also evaluate the running speed (FPS) and calculation amount (MACs) for a fair comparison.

For TuSimple benchmark, the results of FANet along with other state-of-the-art methods are shown in Table 3. Our proposed FANet performs the best in F1 score and also achieves good performance in other metrics. It is clear that the accuracy in TuSimple is relatively saturated, and the accuracy improvement in state-of-the-art methods is also small. Nevertheless, our proposed FANet also performs high accuracy with high efficacy simultaneously. FANet is 33 times faster than SCNN [10], almost 8 times faster than Line-CNN [17], about 3 times faster than ENet-SAD [12], and 2 times faster than PolyLaneNet [16]. Compared with the UFLD [19], although it runs faster than ours, the false positive rate of UFLD is too high, reaching 19.05%, making it difficult to be applied to actual scenarios.

For CULane benchmark, the results of FANet along with other state-of-the-art methods are as shown in Table 4. Our FANet achieves state-of-the-art performance while with high running speed. In crowded, dazzle, no line, cross, and night scenarios, FANet outperforms the other methods. Compared with SIM-CycleGAN [39], although it is specifically designed for different scenarios, FANet is also close to it in many metrics, or even better. Compared with the knowledge distillation method IntRA-KD [14] and the network search method CurveLanes-NAS [40], FANet has a higher F1 score of 3.09% and 4.09% F1 respectively. Compared with UFLD, although it is faster than ours, FANet outperforms it with a 7.09% F1 score.

The visualization results of FANet on TuSimple and CULane are also shown in Figure 6. Although the anchor lines are all straight in FANet, it does not affect the fitting of curved lane lines, as shown in the second row of Figure 6. Besides, FANet also has a strong generalization in various scenarios, such as dazzle, crowded, and night scenes, as shown in the fourth row of Figure 6.

### 4.4. Ablation Study

#### 4.4.1. The Number Setting of Anchor Lines

Efficiency is crucial for a lane detection model. In some cases, it even needs to trade some accuracy to achieve the application’s requirement. In this section, we compare the performance of different numbers of anchor lines settings. In addition to the F1 score, we also compared the running speed (FPS), calculation complexity (MACs), and training time (TT).

As shown in Table 5, until the number of anchor lines is equal to 1000, as the number increases, the F1 score also increases. However, if there are too many anchor lines, i.e., 1250, the F1 score will drop slightly. During the inference phase, the predicted proposals are filtered by non-maximum suppression (NMS), and its running time depends directly on the number of proposals, therefore the number of anchor lines directly affects the running speed of the method.

#### 4.4.2. The Effect of Graph-Based Global Feature Aggregator

As shown in Table 6, under the same backbone network, i.e., ResNet-18, the F1 value is 74.02% when GGFA is not added, and the F1 value is increased to 75.05% after adding GGFA, an increase of 1.03%. The performance improvement shows that our proposed GGFA can effectively capture global information by long-distance weight learning. At the same time, the performance gap also proves the importance of global features with strong perception for lane detection.

GGFA measures the distance between anchor lines by MLP and generates similarity through softmax operation. To better observe the relationship between the various anchor line features, we draw the three most similar anchor lines of the predicted lanes, as shown in Figure 7. It is clear that the anchor lines with big similarities are always close to the predicted lanes. Besides, these anchor lines always focus on some visual cues, e.g., light changes and occlusion, which makes the method can capture some important information.

#### 4.4.3. Performance Comparison under Different Groupings in DFC

As mentioned in Section 3.2, DFC divides the high-dimensional feature into three groups. In this part, we further research the effect of different grouping methods, i.e., block, interval, and random methods. The block method directly divides the high-dimensional features into three groups according to the original arrangement order; the interval method internally takes out the high-dimensional features according to the number of channels, and puts them in three groups; the random method divides the high-dimensional features into three groups after shuffling in the channel dimension.

As shown in Table 7, It is clear that the interval method achieves the best performance, and the random method achieves the second performance, yet the performance of the block method has a large gap. For the feature of each group, the first two methods cover the full range of original channels while the block method only takes the information of 1/3 area. The block method breaks the structure of the original feature, so resulting in an accuracy gap. It also proves the assumption we proposed, i.e., the high-dimensional feature has a great amount of redundant information, the suitable division does not affect its representation ability.

#### 4.4.4. The Effect of Different Channel Dimensions in DFC

To verify the effect of our proposed DFC module, we apply it to high-dimensional feature compression with different dimensions. ResNet-18 and ResNet-50 are used to generate 512 and 2048 dimensions of features, respectively. In ResNet-18, high-dimensional features are compressed from 512 to 64 dimensions while 2048 to 64 in ResNet-50. In terms of task difficulty, it is undoubtedly more difficult to reduce feature dimensions from 2048 to 64.

As shown in Table 8, compared with 1 × 1 convolution, our proposed DFC achieves 0.44% F1 score improvement in ResNet-18, and 0.78% F1 score improvement in ResNet-50. With the same 64-dimensional compressed features, the accuracy is improved significantly after applying our proposed DFC module. It proves that DFC can indeed preserve more information compared with 1 × 1 convolution. Besides, the performance improvement of DFC is more obvious when the feature dimension is higher. It shows that DFC performs better when the dimensional difference is great. At the same time, the consistent improvement in different dimensions also proves the effectiveness and strong generalization of DFC.

## 5. Discussion

According to the results in Table 4, our proposed FANet is not ideal to detect curve lanes. Compared with other scenarios, the results of the curve scene are unsatisfactory. Therefore, we discuss the reason for this phenomenon.

We first make a visualization towards the curve scene in CULane. As shown in Figure 8, it is clear that the predictions are straight. The predictions only find the locations of target lanes and do not fit the curvature of lanes. However, the predictions of TuSimple are indeed curved as shown in Figure 6. Therefore, this is not the problem of the model or code implementation.

We further discuss CULane dataset itself. As shown in Table 9, we count the number of images in various scenarios in CULane. We found that curve lanes are rare in CULane, which are only 1.2% of training images. It means that almost all the lanes in CULane are straight, result in significant bias. It can also explain why the predictions of our model are all straight in CULane.

However, after confirmation, we found that the prediction results in Figure 8 are all regarded as correct because of the high degree of coincidence. Therefore, it can not explain the accuracy gap compared with other methods in the curve scene. Then, we train the model many times and found the experimental results in the curve scene fluctuate greatly. As shown in Table 10, the accuracy gap between the best and the worst is huge, i.e., 2.36 % F1 score. It seems that the few training samples of the curve scene make the accuracy unstable.

## 6. Conclusions

In this paper, we proposed a fast and accurate lane detection method, namely FANet. FANet alleviates three main difficulties of lane detection: (1) how to efficiently extract local lane features with strong discrimination ability; (2) how to effectively capture long-distance visual cues; (3) how to achieve strong real-time ability and generation ability. For the first difficulty, we utilize Line Proposal Unit (LPU) to generate anchor lines over the image with strong shape prior, and efficiently extract local features by their locations. For the second difficulty, we propose a Graph-based Global Feature Aggregator (GGFA), which treats local features as nodes and learns global lane features with strong perception by establishing graph structure. For the third difficulty, our goal is to improve the accuracy of the model without affecting the running speed. Therefore, we propose a Disentangled Feature Compressor (DFC), which is a general and well-designed module for feature compression with a large dimension gap. DFC greatly improves the upper bound of accuracy while without speed delay. We also evaluate the generation of our method in various scenarios, the consistent outstanding performance proves the strong generation ability of FANet.

## Figures and Tables

**Figure 1 sensors-21-04657-f001:**
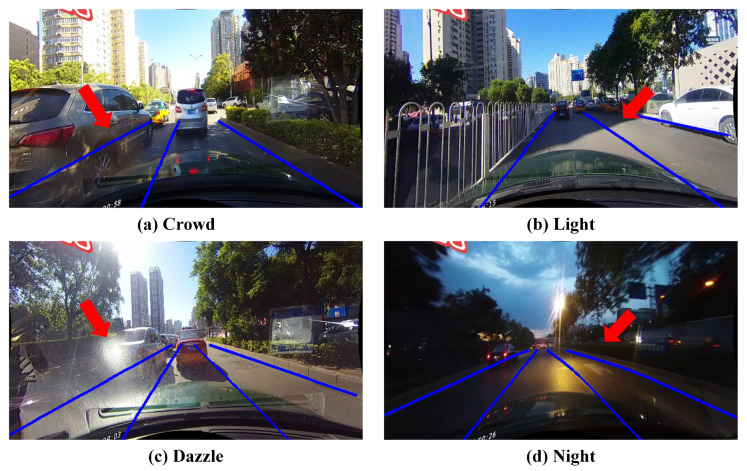
Examples of complex traffic driving scenarios. Some difficult areas are marked with red arrows.

**Figure 2 sensors-21-04657-f002:**
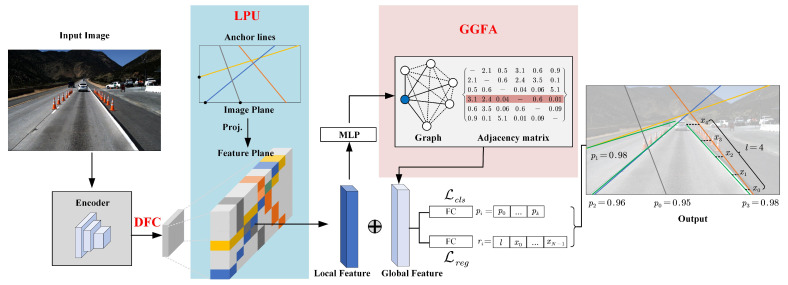
An overview of our method. An encoder generates high-dimensional feature maps from an input image. DFC is then applied to reduce the dimension of features. Subsequently, LPU generates a large number of predefined anchor lines. They are projected onto the feature maps, and the local features can be obtained by pixel-wise extraction. Then, with the input of the local features, GGFA aggregates the global feature with a strong perception. Finally, the concatenated features are fed into two layers (one for classification and another for regression) to make the final predictions.

**Figure 3 sensors-21-04657-f003:**
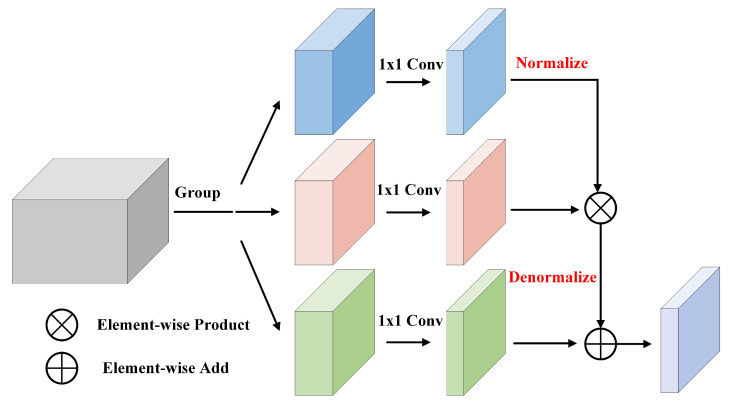
The structure design of DFC. Best viewed in colors.

**Figure 4 sensors-21-04657-f004:**
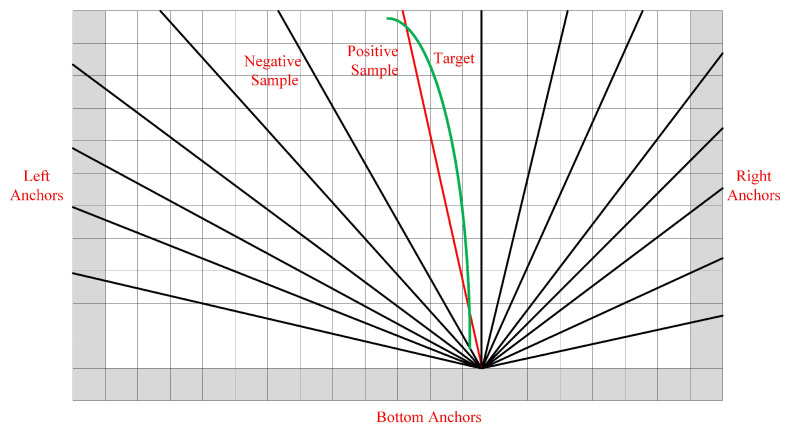
The illustration of anchor line generation process. The green line is the target lane. The red line is the anchor line closest to the target, which is treated as the positive sample. The black lines are negative samples. The gray grids are the starting coordinates for generating the anchor line, including left anchors, bottom anchors, and right anchors.

**Figure 5 sensors-21-04657-f005:**
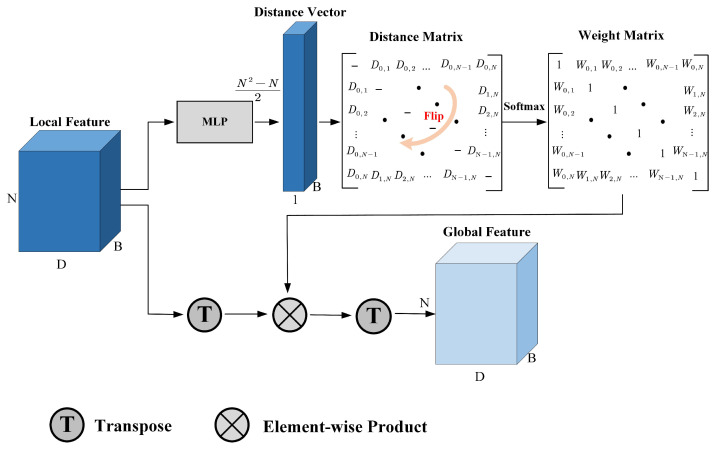
The structure of GGFA. For better visualization, we add the batch dimension *B* to the local feature, performing in a three-dimensional manner. *N* represents the number of pixels in each anchor line, which is equal as $H_{F}$. *D* is the feature dimension, which is equal as $\hat{C}$.

**Figure 6 sensors-21-04657-f006:**
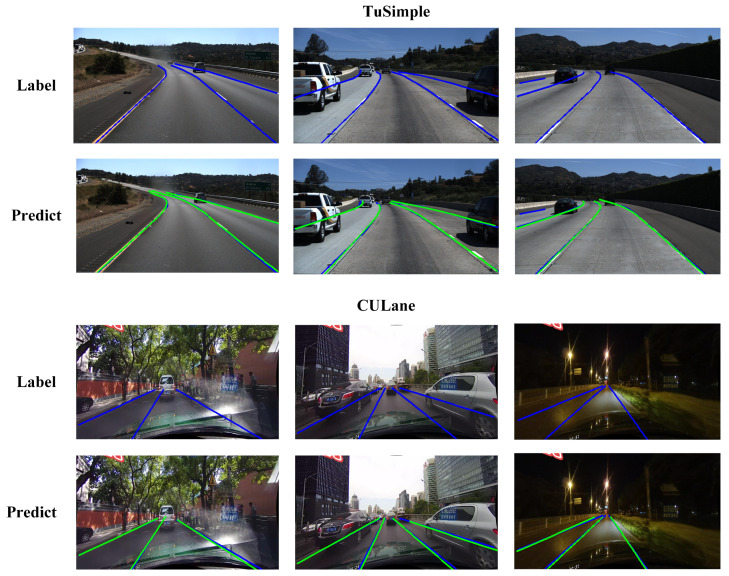
Visualization results on TuSimple (top two rows) and CULane (middle two rows). Blue lines are ground-truth and green lines are predictions.

**Figure 7 sensors-21-04657-f007:**
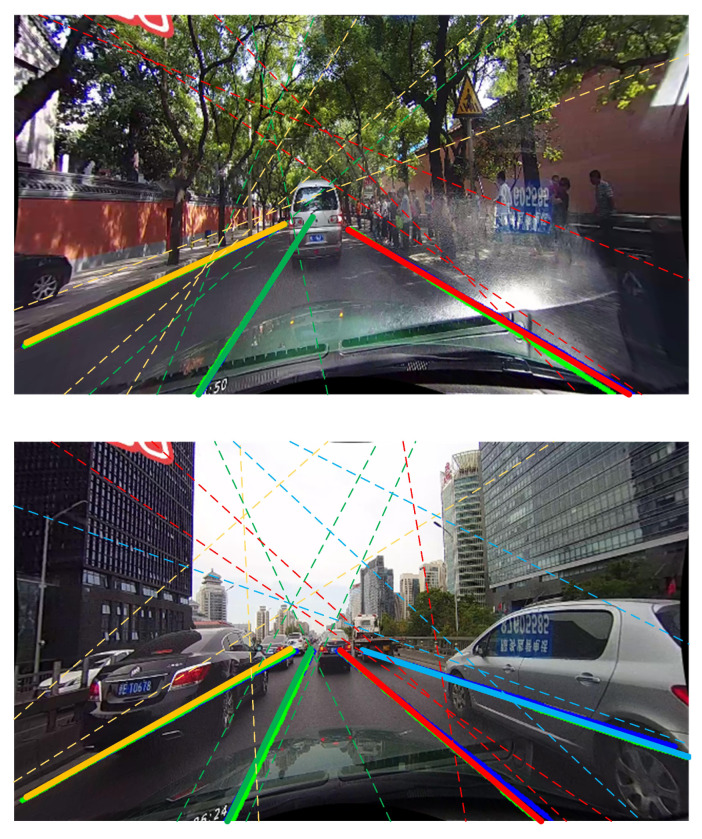
Intermediate visual results of GGFA. The three most similar anchor lines are represented by dashed lines. Different lanes are drawn by different colors.

**Figure 8 sensors-21-04657-f008:**
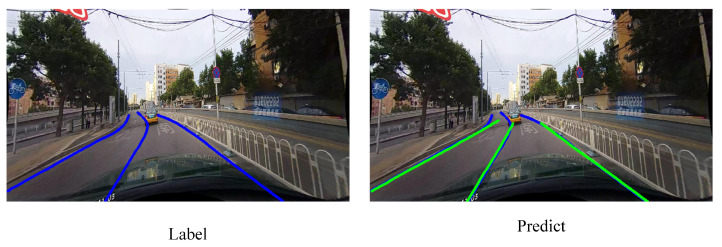
Curve lanes predictions on CULane.

**Table 1 sensors-21-04657-t001:** The angel setting of anchor lines.

Boundary	N	The Set of Angles
left	6	$72^{\circ }$ $60^{\circ }$ $49^{\circ }$ $39^{\circ }$ $30^{\circ }$ $22^{\circ }$
right	6	$108^{\circ }$ $120^{\circ }$ $131^{\circ }$ $141^{\circ }$ $150^{\circ }$ $158^{\circ }$
bottom	15	$165^{\circ }$ $150^{\circ }$ $141^{\circ }$ $131^{\circ }$ $120^{\circ }$ $108^{\circ }$ $100^{\circ }$ $90^{\circ }$ $80^{\circ }$ $72^{\circ }$ $60^{\circ }$ $49^{\circ }$ $39^{\circ }$ $30^{\circ }$ $15^{\circ }$

**Table 2 sensors-21-04657-t002:** Dataset description.

Dataset	#Frame	Train	Validation	Test	Resolution	#Lane	#Scenarios	Environment
TuSimple [24]	6408	3268	358	2782	1280 × 720	≤5	1	highway
CULane [10]	133,235	88,880	9675	34,680	1640 × 590	≤4	9	urban and highway

**Table 3 sensors-21-04657-t003:** State-of-the-art comparisons on TuSimple. For fair comparison, frames per second (FPS) was measured on the same machine used by our method. The best and second-best results across methods are shown in boldface and underlined, respectively. These blank values indicate that the results are not published in their papers, and their codes and models are not available either.

Method	F1 (%)	Acc (%)	FP (%)	FN (%)	FPS	MACs (G)
FastDraw [28]	94.59	94.90	6.10	4.70		
Line-CNN [17]	96.79	**96.87**	4.42	1.97	30.0	
PointLaneNet [20]	95.07	96.34	4.67	5.18	71.0	
E2E-LMD [15]	96.40	96.04	**3.11**	4.09		
SCNN [10]	95.97	96.53	6.17	**1.80**	7.5	
ENet-SAD [12]	95.92	96.64	6.02	2.05	75.0	
UFLD [19]	87.87	95.82	19.05	3.92	**425.0**	
PolyLaneNet [16]	90.62	93.36	9.42	9.33	115.0	**1.7**
**FANet (Ours)**	**96.81**	95.71	3.66	2.86	248.0	9.3

**Table 4 sensors-21-04657-t004:** State-of-the-art comparisons on CULane. For fair comparison, frames per second (FPS) was measured on the same machine used by our method. Because the images in “Cross” scene have no lanes, only false positives are shown. The best and second-best results across methods are shown in boldface and underlined, respectively. These blank values indicate that the results are not published in their papers, and their codes and models are not available either.

Method	Total	Normal	Crowded	Dazzle	Shadow	No line	Arrow	Curve	Cross	Night	FPS	MACs (G)
E2E-LMD [15]	70.80	90.00	69.70	60.20	62.50	43.20	83.20	**70.30**	2296	63.30		
SCNN [10]	71.60	90.60	69.70	58.50	66.90	43.40	84.10	64.40	1990	66.10	7.5	
ENet-SAD [12]	70.80	90.10	68.80	60.20	65.90	41.60	84.00	65.70	1998	66.00	75.0	
IntRA-KD [14]	72.40										100.0	
SIM-CycleGAN [39]	73.90	**91.80**	71.80	66.40	**76.20**	46.10	**87.80**	67.10	2346	69.40		
CurveLanes [40]	71.40	88.30	68.60	63.20	68.00	47.90	82.50	66.00	2817	66.20		**9.0**
UFLD [19]	68.40	87.70	66.00	58.40	62.80	40.20	81.00	57.90	1743	62.10	**425.0**	
**FANet (Ours)**	**75.49**	91.30	**73.45**	**67.22**	70.00	**48.73**	86.36	64.18	**1007**	**69.45**	248.0	9.3

**Table 5 sensors-21-04657-t005:** The efficiency trade-offs of different anchor line numbers on CULane using the ResNet-18 backbone.“TT” represents the training time in hours.

N	F1 (%)	FPS	MACs (G)	TT (h)
250	67.32	281	8.7	4.6
500	73.56	274	8.8	5.3
750	74.10	263	9.1	6.8
1000	75.49	248	9.3	10.2
1250	75.42	231	9.7	10.6

**Table 6 sensors-21-04657-t006:** The effectiveness of GGFA.

Method	Backbone	F1 (%)
w/o GGFA	ResNet-18	74.02
w/ GGFA	ResNet-18	75.05 (+1.03)

**Table 7 sensors-21-04657-t007:** The influence of different grouping methods on DFC. The best result is marked in bold.

Grouping Method	Backbone	F1 (%)
Block	ResNet-18	75.17
Interval	ResNet-18	**75.49**
Random	ResNet-18	75.35

**Table 8 sensors-21-04657-t008:** The effect of different channel dimensions in DFC. 1 × 1 Conv means 1 × 1 convolution.

Method	Dimension	Backbone	F1 (%)
1 × 1 Conv	512	ResNet-18	75.05
DFC	512	ResNet-18	75.49 (+0.44)
1 × 1 Conv	2048	ResNet-50	75.46
DFC	2048	ResNet-18	76.24 (+0.78)

**Table 9 sensors-21-04657-t009:** The image number of different scenarios in CULane.

Normal	Crowded	Dazzle	Shadow	No Line	Arrow	Curve	Cross	Night
9621	8113	486	930	4067	890	422	3122	7029

**Table 10 sensors-21-04657-t010:** The results of multiple experiments on the curve scene in CULane.

1-th	2-th	3-th	4-th	5-th
64.18	63.74	65.53	66.10	64.62

## Data Availability

Not applicable.

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
