# Peer review of "Fast and Accurate Lane Detection via Graph Structure and Disentangled Representation Learning"

_sensors, 2021, doi:10.3390/s21144657_

Round 1

Reviewer 1 Report

Article describes the solution to the problem of lane detection for autonomous vehicles. An efficient and accurate method (FANet architecture) has been  proposed. The architecture is inspired by the best detection solutions (ie YOLOv3, SSD) and classification (ResNet). The feature extractor, compression and global feature aggregator have been combined in an innovative way. The method was tested on two popular benchmarks and compared with a number of reference methods. It has been shown that the proposed solution is the leader in the F1 score metric, with simultaneous quick operation (low computational cost). It is quite likely that a high degree of generalization of prediction (resistance to changes in the external conditions of the scene under study) was obtained, among other things, at the expense of detecting curved stripes, which may be important in specific applications. An interesting, innovative work that makes a significant contribution to the research area.

Comments:

- Why the tables do not include FPS (frames per second)?

- Do the methods without these values ​​make it impossible to measure FPS?

Author Response

We thank you for the careful remarks about our work!

Point 1: Why the tables do not include FPS (frames per second)?

Response 1: We have already listed the results of FPS in Table 3, Table 4, and Table 5. Maybe it’s not obvious enough, we’ve bolded the metrics in Table 3 and Table 4 and added the full name of FPS (i.e., frames per second) in the revised version.

Point 2: Do the methods without these values make it impossible to measure FPS?

Response 2: In table 3 and table 4, there are some values that are empty, because the corresponding metrics are not published in their papers. The code and model of these methods are not publicly available, so FPS and other metrics cannot be tested. In the revised version, we have indicated the reasons in Table 3 and Table 4.

Reviewer 2 Report

The authors proposed fast and accurate lane detection method. It can efficiently extract local lane feature, capture long distance visual cues, and can achieve real-time ability in detection and classification of lane. The used Line Proposal Unit can generate anchor lines over the image with strong shape prior, extracting local feature by their locations.

Detail comments.

  • You wrote in sect. 4.2. “To save memory, the input images are resized to 360 x 640 by bilinear interpolation”. Please explain how this can change results in comparison with other methods.
  • This reviewer saw (Table 3, dataset TuSimple.) that the proposed method resulted in slightly less performance metrics, such as Precision, FP, and FN in comparison with state-of-the art techniques. Only achievement of this authors´ approach is high processing speed FPS except UFLD that presented poor FP value.
  • Please correct in line 217 (page 5): F2, and F3 are analogous to the gamma and betta in eq. (1).
  • Please carefully revise grammar and stylistic of your manuscript, there are several errors, mainly in commas.

Author Response

We thank you for the careful remarks about our work!

Point 1: You wrote in sect. 4.2. “To save memory, the input images are resized to 360 x 640 by bilinear interpolation”. Please explain how this can change results in comparison with other methods.

Response 1: Changing the size of the image is a default setting in lane detection. The conventional settings are 360x640, 288x512, 288x800, etc. Because this is not the focus of this work, we did not deliberately explore the impact of size on accuracy, memory, and speed. Therefore, we have changed the expression of this sentence to “For all datasets, the input images are resized to 360 x 640 by bilinear interpolation during training and testing”.

Point 2: This reviewer saw (Table 3, dataset TuSimple.) that the proposed method resulted in slightly less performance metrics, such as Precision, FP, and FN in comparison with state-of-the art techniques. Only achievement of this authors´ approach is high processing speed FPS except UFLD that presented poor FP value.

Response 2: In addition to the high speed, our method also achieved the best F1 score in Table 3. F1 score is a balance between precision rate and recall rate, which is also important for evaluating the performance of the method. As mentioned in our paper, TuSimple is relatively saturated, and the accuracy improvement in state-of-the-art methods is small. Nevertheless, our proposed FANet also achieves high accuracy with high efficacy simultaneously.

Point 3: Please correct in line 217 (page 5): F2, and F3 are analogous to the gamma and betta in eq. (1).

Response 3: This is a mistake in writing, thank you for pointing it out. We have corrected it in the revised version.

Point 4: Please carefully revise grammar and stylistic of your manuscript, there are several errors, mainly in commas.

Response 4: We have corrected them in the revised version, thank you for your suggestions.